# Strategies and Challenges to Improve Cellular Programming-Based Approaches for Heart Regeneration Therapy

**DOI:** 10.3390/ijms21207662

**Published:** 2020-10-16

**Authors:** Lin Jiang, Jialiang Liang, Wei Huang, Zhichao Wu, Christian Paul, Yigang Wang

**Affiliations:** Department of Pathology and Laboratory Medicine, College of Medicine, University of Cincinnati Medical Center, Cincinnati, OH 45267-0529, USA; jiangl3@ucmail.uc.edu (L.J.); liangjl@ucmail.uc.edu (J.L.); huangwe@ucmail.uc.edu (W.H.); Wuzc@ucmail.uc.edu (Z.W.); paulca@ucmail.uc.edu (C.P.)

**Keywords:** Heart regeneration, induced pluripotent stem cells, cardiac differentiation, direct reprogramming, cardiomyocyte, fibroblasts

## Abstract

Limited adult cardiac cell proliferation after cardiovascular disease, such as heart failure, hampers regeneration, resulting in a major loss of cardiomyocytes (CMs) at the site of injury. Recent studies in cellular reprogramming approaches have provided the opportunity to improve upon previous techniques used to regenerate damaged heart. Using these approaches, new CMs can be regenerated from differentiation of iPSCs (similar to embryonic stem cells), the direct reprogramming of fibroblasts [induced cardiomyocytes (iCMs)], or induced cardiac progenitors. Although these CMs have been shown to functionally repair infarcted heart, advancements in technology are still in the early stages of development in research laboratories. In this review, reprogramming-based approaches for generating CMs are briefly introduced and reviewed, and the challenges (including low efficiency, functional maturity, and safety issues) that hinder further translation of these approaches into a clinical setting are discussed. The creative and combined optimal methods to address these challenges are also summarized, with optimism that further investigation into tissue engineering, cardiac development signaling, and epigenetic mechanisms will help to establish methods that improve cell-reprogramming approaches for heart regeneration.

## 1. Introduction

Ischemic heart disease often leads to congestive heart failure, which is the leading cause of death worldwide. The adult mammalian heart has minimal capacity to generate the contractile cells lost after ischemic injury, and they are replaced with circulatory deficient scar formations as part of the normal heart remodeling process. Recent efforts have been devoted to the development of heart regeneration therapeutics, but stem cell transplantation protocols (such as MSC) have failed to restore structural and functional integrity in affected patients [1]. Encouragingly, induced pluripotent stem cells (iPSCs) can give rise to a number of transplantable cardiac contractile cells to potentially replenish the compromised endogenous cardiomyocyte pool [2]. In addition, patient-specific iPSCs contain genetic information of inherited heart diseases, allowing for a more accurate study of pathological mechanisms [3]. Recent developments in which fibroblasts are directly converted into cells with cardiomyocyte-like phenotypes and functions (direct reprogramming) can be applied to bypass pluripotency, and offer another promising approach for myocardial regeneration. In this review, we provide insights into the development of cardiac regenerative approaches using iPSCs and direct reprogramming of fibroblasts, and conclude with the major advancements in improving cardiomyocyte induction efficiency and maturation.

## 2. Cardiac Generation of iPSCs

Functional cardiovascular cells derived from iPSCs represent a major priority of regenerative medicine research. Methods for cardiac differentiation of iPSCs have become increasingly effective as developmental biology protocols have been optimized. Embryoid body (EB), monolayer culture, and inductive coculture are the three major approaches for the differentiation of iPSCs into cardiomyocytes, and can be achieved via the modification of various signaling factors [4]. There are four major signaling pathways (BMPs, Wnts, TGF-β/Activin/Nodal, and FGFs) involved in the early stages of iPSC differentiation that establish a highly specific temporal window for effectiveness [5,6]. Several groups have reported success in similar cardiac induction in in vivo studies, showing that synergistic relationships exist between growth factors such as BMP4 and FGF2 [7]. Furthermore, the chromatin pattern could be pre-activated at the promoters of cardiac genes. For example, the subunit of *Baf60c*, *WHSC1*, and *SMYD1* are the major epigenetic factors that enable rapid activation of cardiac genes in response to developmental signals [8]. 

Another critical aspect of cardiac regeneration is the generation of desired subtypes in cardiomyocyte differentiation, such as atrial, ventricular, or pacemaker cells. Significant evidence has shown that inhibition of the retinoic acid signal can increase the population of the ventricular subtype, whereas the addition of retinoic acid increases the atrial population (indicating the possible involvement of other signaling pathways) [9,10]. It has also been shown that the epigenetic structure of iPSCs undergoes progressive changes across the genome during differentiation. 

*De novo* transplantation of iPSC-CMs can deliver further benefits. Intramyocardial delivery of iPSC colonies was first established to rescue postischemic myocardial structure and function, achieving in situ regeneration of cardiac muscle, smooth muscle, and endothelial tissue [11]. After injection into the infarcted heart of a macaque, the allogeneic iPSC-derived CMs engrafted and persisted for 12 weeks with evidence of improved cardiac function, demonstrating long-term survival of iPSC-derived CMs while avoiding potential tumor formation [12]. IPSC-derived CMs have also been shown to integrate into infarcted guinea pig heart, improving cardiac function and suppressing arrhythmias [13]. Human iPSC-derived cardiovascular progenitors have been successfully tested more recently in preclinical studies of canine and porcine MI models [14].

## 3. Challenges for Use of iPSCs in Heart Disease Therapy

Many concerns exist regarding the clinical applications and use of iPSC-based cell therapies for cardiac disease. These include (but are not limited to) its immature state of development, poor engraftment rate, inconsistencies in CM differentiation rate, the risk of tumorigenesis, and ultimately, immune rejection. The challenges of iPSC application need to be addressed fully in translational studies before widespread clinical therapies can be optimized.

### 3.1. Improving the Maturation of iPSC-CMs

Although a majority of differentiation methods have been established, it remains difficult to generate fully mature CMs in vitro that possess identical electrophysiological properties and ultrastructural features to their native adult counterpart [15,16]. These generated cells more closely resemble CM in the fetal stages of development with a lower maximum contractile force, reduced calcium stores and cycling, slower upstroke velocity, and immature mitochondrial function [17]. Unfortunately, these defects in electrophysiological iPSC-CM maturation result in arrhythmogenic risk in cell replacement therapy [18]. The ectopic automaticity of immature iPSC-CM would be uncoupled with the rhythm of host’s myocardium and contribute to the arrhythmogenic propensity after transplantation [19].

Cardiomyocytes require exposure to a myriad of biophysical and biochemical factors in vivo. Standard culture conditions lack critical factors to mimic the native CM microenvironment, limiting the process of CM maturation. In addition to long-term culture [20] findings, several biomaterial-engineering studies have shown that a stiffness matrix and physiological cell substrate significantly improve the structural and functional maturation of iPSC-CMs [21,22]. Jacot et al. cultured rat ventricular myocytes on collagen-coated polyacrylamide gels with elastic moduli of 10 kPa that produced aligned sarcomeres and great contractile stress [23]. Similarly, Hazeltine et al. found that polyacrylamide hydrogels embedded with elastic moduli ranging from 4.4 to 99.7 kPa increased the contraction stress of all types of human iPSC-CMs [24]. The addition of polypyrrole chitosan (PPC) to hydrogel also improved electrical propagation between clusters of iPSC-CMs [25]. Several other studies have revealed that micropatterned substrates adopting a rod-shaped morphology improved the sarcomere organization and electrophysiological function of CMs [26,27]. Human iPSC-CMs that were patterned with Polydimethylsiloxane (PDMS) channels and exposed to cyclic uniaxial stretch showed excellent mechanical properties in aligned cardiac tissue with enhanced Z-band/Z-body formation and expression of key maturation markers (such as *cTnT*, *MLC2v*, and *MLC2a*) [28]. 

In addition to the matrix types and patterns available for CM attachments, another promising approach involves engineering highly controlled three-dimensional (3D) culture systems. Notably, 3D systems have been the most successful method to boost maturation in vitro that mimics native CM architecture to facilitate cell-cell and cell-matrix interactions [29]. Ashley et al. created a 3D cardiac extra cellular matrix (ECM) scaffold that possessed increased calcium amplitude and kinetics in the sarcoplasmic reticulum [30]. Sean’s team generated symmetric 3D constructs that enable rapid aggregation of human iPSC-CMs by use of faraday waves, exhibiting more mature characteristics including greater levels of contractile stress, beat frequency, and contraction-relaxation rates [31]. In addition, a PDMS cell-culture substrate with 3D topography was developed that accelerates both human iPSC-CM differentiation and maturation processes via reorganization of the cytoskeletal network [32]. Furthermore, it has been demonstrated that the modulation of mechanical 3D tissue stress has a direct effect on CM function and maturation. Abilez’s group found that human engineered heart muscle (EHM) was subjected to a passive 7 cm stretch by calculating the distinct stress distributions across EHMs, which showed considerably increased maturation gene expression (particular in T-tubule protein) and calcium amplitude through iPSC to CM [33]. These results indicate the potential role of a 3D system combined with patterned arrangement in iPSC-CMs maturation, even though the mechanism of iPSC-CM maturation is yet to be elucidated.

Deficiencies in electrical signals and ion currents (such as sodium (INa), potassium (IK1 and IKr), L-type, and T-type Ca2^+^ channel) are also common in newly derived, immature CMs [34]. To transit iPSC-CMs into a mature electrical state, Nunes and colleagues used electrical field stimulation of increased pacing from 1 Hz to 6 Hz over 1 week on human PSC-CMs, which induced multiple aspects of maturation concurrent with elongated morphology in cell-cell junctions and numbers of mitochondria, advanced conduction velocity, and calcium handling in calcium stores and cycling [35]. Similar results have been reported in 3D systems, with beneficial effects that persisted two weeks after electrical stimulation was removed [36]. A recent study found that using a ramped electrical stimulation scheme (modulated pacing frequency from 2 Hz to 6Hz by increase of 0.33Hz per day over 2 weeks, followed by 1 week at 2 Hz) enhanced the structural maturity of iPSC-CMs. Stimulation resulted in CMs with adult-like gene expression profiles that were closely analogous with *in vivo* myocardium, except for their membrane potential [16]. Notably, Liu’s team designed 3D self-organized tissue rings (SOTRs) to control beating frequency by adjusting the ReWs waves (which has long been investigated as a model for arrhythmia studies), making a CM beat robustly at 2–4 Hz for 89 days [37]. This mathematical model serves as a practical system for the origination and propagation of action potential for future production of mature iPSC-CMs. 

In addition to maturation induced by biophysical stimuli, metabolic regulation and biochemical stimuli have emerged as key components that control iPSC-CM maturation. Among these, thyroid hormone T3 exerted a broad impact on CM maturation, including contractile force, Ca^+^ handling, and mitochondrial respiration [38]. Human iPSC-CMs treated with 20 ng/mL of T3 showed obvious morphological differences in larger cell sizes and increased sarcomere length [39]. Similar to T3, glucocorticoid modulation has shown promise in improving oxygen consumption, ion-channel levels, and contractility of iPSC-CMs [40]. Although T3 treatment alone is insufficient to transform CMs into an adult-like state, a combined approach using T3 with glucocorticoid can synergistically induce extensive T-tubule formation and calcium kinetics [41]. Other biochemical factors like insulin-like growth factor 1 (IGF1) and neuregulin 1β (NRG1) and their receptors can regulate cardiac metabolism and contractility through the RAS-RAF-MEK-ERK, PI3K-AKT-mTOR, or PLC-IP3-DAG pathways [42]. For example, a small molecule Torin 1 was used to transiently inhibit mTOR pathway that enhanced maturation of iPSC-CMs, whereas deletion of IGF1 and INS receptors resulted in disrupted sarcomere and mitochondrial morphology [43]. Human iPSC-CMs showed remarkable progression in fiber alignment and sarcomere length using the Wnt signaling modulation protocol during a 120-day culture period [44]. While Sean Wu and colleagues found that the Wnt signaling pathway activation, combined with removal of cell–cell contact, achieved scalable human iPSC-CMs with comparable contractility [45]. These results suggest the value of an integrated approach using tissue engineering and signaling pathway regulation in future cardiac regenerative strategies. 

Energy metabolism and oxygen tension are other environmental cues critical to CM maturation. Metabolism during iPSC-CMs generation is switched from glycolysis to fatty acid oxidation to facilitate CM maturation. Horikoshi and Correia both showed that the maturation of human iPSC-CMs was enhanced by the addition of a fatty acid-rich, glucose-free medium, with a specific improvement in activated oxidative phosphorylation, dense mitochondria with lamellar cristae, and increase binucleation [46,47]. They further identified the high levels of gene expression encoding ion channels, cardiac ryanodine receptor RYR2, and key metabolic transcription regulator peroxisome proliferator-activator receptor PPARα in these cultured iPSC-CMs. A comparable study reported by Feyen et al. showed similar results [48] when they developed a low glucose and high oxidative substrate medium to induce maturation of human iPSC-CMs using a 3D culture system. The metabolically matured CMs showed increased Na^+^ channels and sarcoplasmic reticulum Ca^2+^ cycling dependency that contributed to force generation. When human iPSC-CMs were subjected to a high glucose environment, HIF1α (hypoxia-induced factor 1α) was activated as a CM maturation blocker, suppressing oxidative phosphorylation [49]. Conversely, increased oxygen tension inhibited HIF1α signaling, improved the metabolism of human iPSC-CM, and played an important role in the mechanism of CM maturation [50]. 

Human iPSC-CMs are used in modeling inherited cardiac diseases for numerous translational applications, and make it possible to investigate their pathology for the discovery of novel therapeutic agents. Currently, many current disease models have difficulties in recapitulating disease phenotypes due to the inherent immaturity problem [3]. There has been exciting progress in recent maturation strategies of iPSC-CM, including in the engineering of biophysical cues, hormone intervention, regulation of signaling pathway, and other metabolism manipulations (Figure 1). The physiological contractile force and upstroke velocity are still lower than those of the adult myocardium, and the molecular mechanism of stimuli cues in cardiac maturation has yet to be elucidated. Therefore, the ideal solutions tend to develop combined stimuli methods to manipulate CM maturation, thereby addressing the scalable and functional demand for clinical use [51,52].

### 3.2. Enhancing Engraftment of iPSC-CM

Engrafted cells have a very limited clinical survival rate (<10%), and the long-term retention of CM transplants has become a key challenge when designing CM therapeutics [53]. Ample evidence exists for improved engraftment, survival, and effectiveness of transplanted CMs using a variety of technological advancements, especially using tissue engineering platforms (cell sheet, patches or scaffolds) [54]. When implanted in an MI model, cardiac tissue sheets (assembled with defined cardiovascular cells derived from iPSCs) show significant and sustained improvement of cardiac function accompanied by neovascularization [55]. Lee et al. developed a collagen-dendrimer biomaterial crosslink with prosurvival peptide analogs that significantly increased cell engraftment in the infarcted area [56]. Moreover, the CMs differentiated from iPSCs were grown on decellularized heart scaffolds to form heart tissues that exhibited spontaneous contractions and generated mechanical force [57]. Laura et al. found that the combination of human iPSC-CMs with biomimetic microparticles (MPs) enhanced long-term survival for two months, with corresponding improvement in heart function and resistance to adverse ventricular remodeling [58]. Similar beneficial effects were also achieved using a 3D printed ECM scaffold [59]. 

Increasing engraftment of iPSC-derived CMs also provides cardioprotective benefits to compensate for damaged heart function through paracrine effects. Several factors, such as cytokines, miRNA, and exosomes are responsible, and potentially vital, for CM regeneration. Transplanted iPSC-derived vascular cells can secrete stimulatory cytokines to instigate an innate regenerative response and induce angiogenesis [60,61]. Cardiac over-expression of miR-15, miR-199a, or miR-590 also has been implicated in paracrine signaling to regulate cell cycle and decrease fibrosis in the infarcted tissue [62]. 

Engineered tissue transplantation is a novel solution for several of the common pitfalls encountered using stem cell therapy in conventional injection/infusion models such as poor cell retention, engraftment quality, and survival of implants [63]. Given the remuscularization principle and ventricular wall augmentation strategy [64], tissue engineering can play a significant role in iPSC-CM replacement therapy in a wide range of heart diseases.

### 3.3. Immune Reaction of iPSC-CM 

Many novel studies have focused on the interaction of transplanted iPSC-CMs with the immune system under MI conditions. Allogenic strategies that avoid adverse immune reactions are considered highly important for the regeneration of iPSC-CMs and subsequent healing process in heart repair due to the time and cost pitfalls of using autologous iPSC-CM replacement [65]. Conceptually, HLA (human leukocyte antigens) matching or HLA removal are the major strategies to evade immune reactions [19]. By using the CRISPR-Cas9 gene editing technique, Akisu’s group developed an advanced HLA matching in iPSCs (HLA-C retained, wherein HLA-A and HLA-B have been bi-allelically deleted) that greatly enhanced immune compatibility via suppressing T and NK cell activity [66]. Besides the retention of HLA-C, the expression of immunomodulatory factors PD-L1 and HLA-G could promote the immune tolerance of iPSC transplants [67,68]. A recent study demonstrated that reduced activity of *CCR2*^+^ macrophages would be beneficial in acute ischemic conditions, while prolonged *CCR2* inhibition resulted in the cardiac functional improvement which diminished after CM transplantation [69]. Thus, differentiated CMs are not sufficient to be adopted widely as a complete therapeutic strategy for heart disease due to interactions between transplanted iPSC-CMs and the immune system. Further studies are needed to clarify the immune effects and increase the efficiency of iPSC-CM engraftment. 

Researchers continue to refine current protocols to improve direct differentiation, maturation of iPSCs-CMs, and cell retention rate. Developmental biology research has focused on optimizing devices, culture conditions, materials, and timing to better recapitulate the myocardium environment in vivo [4,70]. While the use of iPSCs-CMs in clinical therapeutics is still far off, near-term potential may lie in applications that serve as a platform for heart disease modeling and personalized medicine [71].

## 4. Direct Cardiac Reprogramming for Heart Regeneration

### 4.1. Direct Reprogramming of Fibroblast into CMs

The recent development of cell-free, cardiac-fate direct reprogramming methods offers a dual advantage of reducing scar tissue while simultaneously generating new functional CMs without passing through a pluripotent stage [72]. Unlike iPSCs, direct reprogramming methods avoid the major pitfalls of iPSCs techniques including time-consuming cell conversion, the risk of genetic abnormalities, and tumorigenesis [73,74]. Moreover, converting resident cardiac fibroblasts into functional CMs in situ via nonviral, direct reprogramming is another critical advantage that offers great potential for facilitating clinical-based regenerative applications. 

Leda et al. discovered a specific combination of transcriptional factors (TFs) including *Gata4*, *Mef2c*, and *Tbx5* (GMT) that was necessary and sufficient to transform mouse *Thy1*^+^ dermal or cardiac fibroblasts into induced CMs (iCMs) [75]. The iCMs proved to exhibit similar global gene expression, epigenetic imprinting, and well-defined sarcomeric structures which were similar to endogenous CMs. Subsequently, Song et al. screened conserved cardiac lineage TFs to identify a cocktail of *Gata4*, *Mef2c*, *Tbx5*, and *Hand2* (GMTH) that generated nearly five-fold *cTnT^+^/αMHC-GFP^+^* iCMs as compared to the GMT combination [76]. This finding suggests that the presence of *Hand2* enhances the efficiency of cardiac reprogramming. 

### 4.2. Challenges for Translating the Direct Reprogramming Approach

#### 4.2.1. Increasing Conversion Efficiency

Nam et al. reported that about 10–30% of GHMT-induced fibroblasts express *cTnT*, whereas nearly 1% of cells formed sarcomere structures that correlated with spontaneous beating activity (0.16%) [77]. Several research groups have developed strategies to improve cardiac reprogramming efficiency through the use of differential TF combinations. Christoforou et al. further demonstrated a TF cocktail including *Myocd*, *SRF*, *Mesp1*, or *SMARCD3* with the GMT that enhanced *cTnT* mRNA expression [78]. Addie et al. employed a calcium reporter to screen candidate factors and found that the addition of *Nkx2.5* to GMTH cocktail maximized reprogramming efficiency, showing more than 50-fold robust calcium oscillation and spontaneous beating [79]. The candidate-enhancing TFs, such as *Ankrd1*, *Gata6*, *Mef2A*, *Myocd*, *Ppargc1a*, *Tbx20*, *Tead4*, and *Wt1*, have also been predicted in silico by using computational methods and constructing gene regulatory networks [80]. In addition, Wang et al. found that isoform2 of *Mef2C* (Mi2) can induce higher reprogramming efficiency than its isoform4 (Mi4) in fibroblasts, and that polycistronic Mi2-GT retrovirus further promoted successful reprogramming for mature CMs, which was consistent with results reported by Hirai et al. and Abad et al. [81,82,83].

A dose-spatial-temporal balance of the TFs is required to regulate cardiac lineage commitment during heart development. To determine the stoichiometry of reprogramming TFs, Wang et al. investigated the optimal ratio of GMT by using polycistronic constructs with self-cleaving 2A sequences [84]. They showed that a MGT vector (conferred high *Mef2c* expression and low levels of *Gata4* and *Tbx5*) significantly increased reprogramming efficiency of iCMs [85]. Moreover, Liu et al. systematically demonstrated the effect of 2A cleavage peptide on transgenic expression efficiency [86]. They found that the ratio of GMT proteins was crucial for efficient iCM reprogramming. However, Hirai et al. found that there were no beneficial effects conferred using various doses of monocistronic G/M/T/H on cardiac reprogramming of MEFs [82]. Recently, Zhang et al. constructed a quadcistronic vector that induced relatively low *Hand2* and high *Mef2c* protein levels and increased sarcomeric organization, calcium flux, and spontaneous contraction of iCMs [87]. These reports emphasize the importance of stoichiometric expression of TFs and provide insights to facilitate efficient cardiac reprogramming.

Multiple studies have shown that the modification of signaling pathways, the optimization of small molecules or miRNAs, and epigenetic alterations represent potential strategies to promote the efficiency and maturation of direct cardiac reprogramming. GMTH-based conversion of mouse fibroblasts was enhanced by Notch signaling inhibition with DAPT ((S)-tert-butyl 2-((S)-2-(2-(3,5-difluorophenyl)acetamido)propanamido)-2-phenylacetate), resulting in CM reprogramming efficiency up to 70% [83]. The co-administration of AKT serine/threonine kinase 1 (AKT1) with GMTH led to increased CM maturation, evoking contractility in 50% of starting MEFs [83]. Mechanistically, activation of IGF1/PI3K/AKT1 signal pathway regulated downstream gene targets related to TOR activation and Foxo3a inhibition, which played a role in CM maturation [88]. Recently, Zhou et al. found that the zinc finger transcription factor 281 (ZNF381) acted as a nexus of cardiac and inflammatory gene programming, and stimulated the cardiogenic activity of AGHMT in adult mouse fibroblasts [89]. Protein kinase D1 (PRKD1), PKA, PKC, and calmodulin-dependent protein kinases (CAMK) have also been reported as potential enhancers of GMT-based conversion [90]. Consistent with this, Yamakawa et al. found that treatment of serum-free medium containing FGF2, FGF10, and VEGF achieved a faster CM maturation by activating the p38 mitogen-activated protein kinase (MAPK) and PI3K/AKT1 pathway, making *Gata4* dispensable from the GMT cocktail [91]. 

There is increasing evidence that the inhibition of profibrotic signaling pathways promotes reprogramming efficiency. TGF-β signaling behaves as a repressor for activating the transformation of cardiac microvascular endothelial cells, and contributes to cardiac fibrosis. Ifkovits et al. showed that the inhibition of TGF-β signaling (commonly used inhibitor, SB431542) led to a five-fold increase in reprogramming efficiency via GMTHN, and generated more beating iCMs from MEFs and adult cardiac fibroblasts by silencing fibroblast signatures [92]. Zhao et al. identified another TGF-β inhibitor, A8301, that blocked TGF-β receptors and enhanced iCM induction [93]. They further found that inhibition of the TGF-β and Rho-associated kinase (ROCK) pathways converted fibroblasts into functional iCMs with an efficiency up to 60% by forced expression of GMT or GHMT [93]. Furthermore, a combination of TGF-β inhibitor and Wnt inhibitor could significantly augment GMT reprogramming, as evidenced by shortened timing and enhanced quality of iCM conversion [94]. The targeted knockdown of fibroblast-specific TFs such as odd skipped related TF1 (*OSR1*), paired related homeobox (*PRRX1*), and LIM homeobox 9 (*LHX9*) also attenuated the fibroblast gene network to improve cardiac reprogramming [95,96]. 

Suppression of inflammatory signaling has recently been reported to directly improve reprogramming. Muraoka et al. found diclofenac (a nonsteroidal anti-inflammatory drug) greatly enhanced cardiac reprogramming, in part by inhibiting cyclooxygenase-2, prostaglandin E2/prostaglandin E receptor 4, CAMP/PKA, and interleukin 1B signaling in neonatal and adult mouse fibroblasts [97]. The inhibition of C-C chemoking signaling in MEFs or neonatal fibroblasts was also shown to improve cardiac conversion. These results emphasized the involvement of immune response suppression and fibroblast program silencing in cardiac reprogramming progression [98]. The modulation of signaling cascades, therefore, has a positive effect on iCM reprogramming in addition to cardio-induction and cardio-supportive roles.

MicroRNAs (miR) play an important role in cardiac development and CM fate by regulating various genes related to TFs, signaling pathways, and epigenetic modifiers. Jayawawrdena et al. reported that a combination of muscle specific mi-RNAs (miR-1, miR-133, miR-208, and miR-499) with a chemical JAK inhibitor (Janus kinase) was capable of generating functional iCMs from CFs [99]. Subsequently, Muraoka’s group reported that the addition of miR-133 led to a seven-fold improvement in the efficiency of the GMT-induced cardiac reprogramming, and noticeably enhanced the speed of CM maturation by directly targeting Snail1, a master regulator of epithelial-to-mesenchymal transition [100]. The combination of miR-1and miR-133 with GMTH cocktail generated significantly more matured iCMs [93,100]. Supplementing the viral GMT, GMTH, or GMTHMyMe cocktail with a TGF-β inhibitor or synthetic mimics of miR-1, miR-133, or miR-590 increased the CM reprogramming efficiency and speed of mouse, pig, or human cells, which provided new insight into the molecular mechanism of cardiac reprogramming [80,101,102].

#### 4.2.2. Uncovering the Mechanisms of Direct Cardiac Reprogramming

Regulatory epigenetic factors play important roles in determining cell type-specific gene expression during the reprogramming of iCMs. Revealing epigenetic processes helps to better understand the molecular mechanisms underlying iCMs generation. Evidence indicates that epigenomic and transcriptional changes occur within the first 1–2 days of CM reprogramming, followed by a dynamic alternation of histone modifications [103]. Zhao et al. analyzed the epigenetic states of fibroblasts transduced with GHMT and found a peak of H3K4me2 (histone H3 lysine 4 dimethylation) at reprogramming day 7, and the studies of epigenetic dynamics showed that H3K27me3 was reduced, but H3K4me3 was increased in cardiac promoters as early as 3 days of induced reprogramming process [104]. In 2016, Qian et al. discovered Bmi1 (a component of the Polycomb repressive complex I) tobe a major barrier of CM reprogramming by a shRNA screening. The suppression of Bmi1 was associated with upregulated H3K4me3 and downregulated H2K199ub at cardiogenic loci, including *Gata4*, *Nkx2.5*, *Pitx2*, *Tbx20*, and *Hand 2*, thereby enhancing reprogramming efficiency. This also indicated a dispensable role *Gata4* by using Bmi1 shRNA [105]. In 2017, Liu et al. found that the efficiency of CM conversion was improved by targeting Men1, a component of H3K4 methyltransferase *Mll1*. Furthermore, exposure to Ezh2 (enhancer of Zeste homolog2) inhibitor GSK126 and G9a inhibitor Unc0638 also led to an increased reprogramming efficiency via H3K27me3 and H3K9me2 modifications, respectively [106]. In 2019, Qian’s group applied single-cell RNA sequencing to reveal immune-response-associated DNA methylation and downstream targets of TFs that are required for human iCM induction [107]. Hisayuki et al. also reported that GHMT synergistically activated cardiogenic stage-specific enhancers, and highlighted the *Mef2c* binding sites by genome-wide analyse. They found that *Hand2* and AKT1 signaling coordinates the recruitment of other TFs to the enhancer site [108]. These results show that there is a trajectory toward CM fate in fibroblasts after induction by early TF expression, which was associated with epigenetic modifiers targeting a broad landscape of cardiac enhancers and key signaling pathways.

#### 4.2.3. Safety Issues of Gene Therapy

Most iCM generation techniques employ the use of virus-based strategies to deliver reprogramming factors, which raises safety concerns including endogenous gene disruption and insertional mutagenesis [109]. Investigators have tried to avoid this pitfall via nongenetic strategies that employ the use of small molecules to induce reprogramming [110]. In 2014, Ding et al. established their CASD (cell-activation and signaling-directed) approach and achieved successful reprogramming from MEFs and TTFs into iCMs by using a single TF-Oct4 and the four compounds SB431542 (AKT4/5/7 inhibitor), CHIR99021 (GSK-3 inhibitor), Parnate (LSD1/KDM1 inhibitor), and Forskolin (adenylyl cyclase activator) [111]. This finding suggested the feasibility of using small molecules to accurately specify cardiac lineage. In 2015, a direct conversion of mouse fibroblast to CMs was induced using a combination of chemical cocktails CRFVPT (C: CHIR99021, R: RepSox, F: Forskolin, V: Vpa, P: Parnate, T: TTNPB) [112]. These chemical induced CMs (CiCMs) showed similar electrophysiological activities to CMs including spontaneous contraction, calcium transients, and response to adrenergic and muscarinergic stimulants [112]. Conversely, the generation process of CRFVPT-induced CiCMs passed through a cardiac progenitor stage as evidence by the upregulated progenitor markers such as *Sca-1*, *Abcg2*, *Wt1*, *Flk1*, and *Mesp1* at early stage of reprogramming. In 2016, Cao et al. achieved the same direct reprogramming results in a human cell model using a combination of seven small molecules {CHIR99021, A83-01 (TGF-β1 inhibitor), BIX01294 (a histone methyl-transferase inhibitor), AS8351, SC1 (ERK1 inhibitor), Y27632 (ROCK inhibitor), and OAC2 (Oct4-activating compound 2)} [113]. They also found that addition of SU16F (PDGF-β inhibitor) and JNJ10198409 (PDGF receptor tyrosine kinase inhibitor) to the seven molecules increased efficiency by 6–7% at day 30 [113]. The reprogrammed CiCMs passed through a progenitor stage rather than a pluripotent stage and showed contracted features resembling human CMs. All these studies suggest different epigenetic mechanisms of CM reprogramming by using chemical cocktails that induce fibroblasts into a plastic progenitor state towards cardiac lineage differentiation.

The application of bio-safe delivery vectors is another promising method for integration-free iCM reprogramming. For instance, adeno-associated virus (AAV) has been found to optimize expression of multiple TFs for reprogramming [114]. Kazutaka et al. developed a polycistronic Sendai virus expressing GMT (Sev-GMT) that efficiently and rapidly reprogrammed both mouse and human fibroblasts into beating iCMs [115]. Likewise, the nonintegrating adenoviral vector also served as an efficient delivery method in a rat model of reprogramming [116]. Although further optimization is still needed, safer vectors or biomaterials will play a vital role in local delivery and address the safety concerns associated with cardiac gene-mediated reprogramming. 

#### 4.2.4. Enhance Human Cardiac Reprogramming

Despite encouraging results and numerous ideas for potential improvements in regenerative medicine using direct cardiac reprogramming, several challenges remain to be addressed prior to widespread clinical application. Cardiac reprogramming in human fibroblasts is still inefficient and varies between research laboratories. Although contributing factors have been identified in mouse cardiac reprogramming, human iCMs reprogramming failed to generate robust beating in vitro, indicating that the majority of reprogrammed cells were neither sufficient nor highly mature. Applying genetic and epigenetic assays such as single-cell RNA sequencing, chromatin assay, lncRNAs, Cas9-based genome editing technology, protein interaction network mapping, metabolite stimulating, and inflammatory response inhibition will all be helpful to investigate the more detailed mechanisms underlying iCM reprogramming and give rise to better reprogramming efficiency and iCM maturation [107,117]. Similar to iPSC-CMs, the optimization of culture microenvironments, including the extracellular matrix, paracrine factors, contractile forces, and electrical currents, also improved the maturation of newly reprogrammed iCMs. In addition, nonviral or chemical protocols for iCM reprogramming might overcome safety concerns, which could be developed as a promising local delivery method for clinical use. Cardiac reprogramming studies in large animal models will be the next step for future studies.

## 5. Emergence of Cardiac Progenitor Reprogramming 

Recent research has shown that induced cardiovascular progenitor cells (iCPCs) reprogrammed from somatic fibroblasts can serve as a viable source of three cardiovascular lineages that display cardiac-mesoderm-restricted, multipotency and prompt heart generation. In 2012, Islas et al. first reported the conversion of normal human fibroblasts into *Nkx2.5-tdTomato^+^KDR^+^* iCPCs by the overexpression of *Mesp1* and *Ets2* in the presence of Activin A and BMP2, but indicated no sign of endothelial cell (EC) or smooth muscle cell (SMC) differentiation potential [118]. In 2015, Pratico et al. generated *c-Kit^+^* iCPCs from adult human dermal fibroblasts by repeated GMHT mRNA treatments, showing differentiated *cTnT^+^* noncontractile CMs [119]. Instead, Li et al. demonstrated that GHMT protein transduction in combination with BMP4, Activin A, and bFGF allowed for rapid reprogramming from human dermal fibroblasts into iCPC with tri-lineage cardiovascular differentiation potential [120]. These achievements revealed the induction of multipotent iCPCs towards a relative mature state. 

In order to extend the applicability and regenerative capacity of iCPCs, a combination of five TFs (*Mesp1*, *Tbx5*, *Gata4*, *Nkx2.5*, and *Baf60c*) followed by exposure of Wnt activator BIO and JAK/STAT activator LIF was sufficient to reprogram adult mouse fibroblasts into expandable iCPCs [121]. These iCPCs integrated into the embryonic heart tube, differentiated into functional CMs, and contributed to vasculature in the infarcted area by differentiated SMCs and ECs. Additionally, Ding’s group used transient OSKM (*Oct4*, *Sox2*, *Klf4*, and *c-myc*) overexpression associated with the BASC cocktail (BMP4, Activin A, SU5402, CHIR99021) method that successfully gave rise to *KDR^+^/PDGFRA^+^* expandable cardiovascular progenitor cells, resulting in attenuated scar area and improved heart function [122]. However, the precise calculation of iCPC final differentiation was hard to control, with the pitfalls of contamination and the inclusion of noncardiac lineage cells. Panciera et al. found an easy way for the transient expression of YAP or TAZ to convert different types of somatic cells to corresponding tissue-specific progenitor cells, sharing similar functional properties like differentiation ability and self-renewal, both in vitro and *in vivo* [123]. Therefore, reprogramming iCPCs may provide a source of cardiac-specific linage cells that are uncompromised by growth arrest for individualized autologous cell therapies.

## 6. Conclusions

CM generation is a delicate and dynamic process of cardiac lineage programming differentiation. The regeneration of new CMs is regulated by a unique set of transcriptional factors, signaling pathways, miRNA, and epigenetic networks. With defined differentiation and purification protocols, iPSCs-derived CMs offer a safer and more efficient approach and provide a critical tool in vitro for modeling numerous genetically heritable heart diseases and their related molecular mechanisms. The direct reprogramming of iCMs or iCPCs from fibroblasts also provides new, attractive strategies for efficient heart regeneration (Figure 2). 

Despite the various enhancement approaches that have been developed to improve CM induction, cardiac programming and direct reprogramming are still clearly underdeveloped assets that need further laboratory research. Low efficiency, functional immaturity, and safety issues are the primary challenges of heart regenerative approaches, and further investigation in tissue engineering, cardiac development signaling, and epigenetic mechanisms will help to establish methods to address these challenges and further accelerate cell reprogramming-based approaches from bench to bedside.

## Figures and Tables

**Figure 1 ijms-21-07662-f001:**
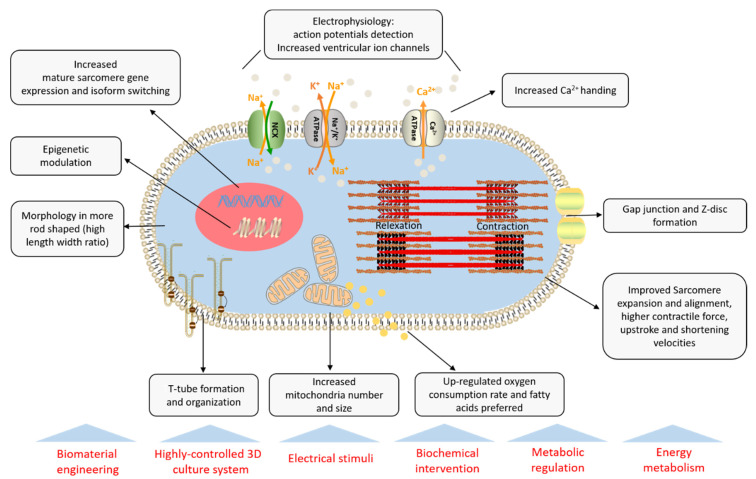
Properties of cardiomyocytes derived from induced pluripotent stem cells. IPSC-CMs feature high expression of key maturation markers, primed ultrastructural morphology, and functional electrophysiology (especially ion channel and contractility), and fatty acid-dependent rather than glycolysis. Current strategies for iPSC-CM production include engineered substrates, electrical stimulation, mechanical intervention, and metabolic modulation with optimized culture conditions.

**Figure 2 ijms-21-07662-f002:**
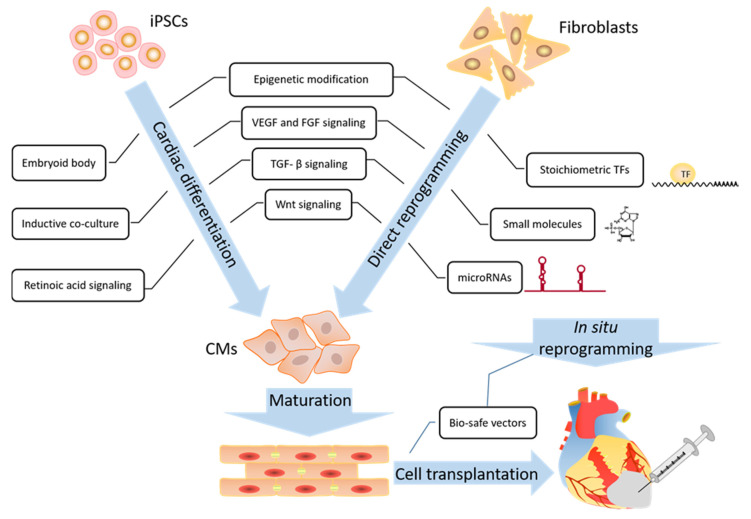
Current developments in cellular programming-based approaches for heart generation therapy. Prematured cardiomyocytes can be generated using both iPSC programed differentiation (left side) and direct reprogramming from various types of fibroblasts (right side), which share similar signaling pathways (such as VEGF, TGF- β, Wnt) and epigenetic modulation to promote conversion efficiency. Other current techniques of direct reprogramming employ transcriptional factors, microRNAs, and small molecules (or a combination of all 3). These have been shown to facilitate direct reprogramming in vitro and in situ for cardiomyocyte transplantation.

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
