# Peer review of "Strategies and Challenges to Improve Cellular Programming-Based Approaches for Heart Regeneration Therapy"

_ijms, 2020, doi:10.3390/ijms21207662_

Round 1

Reviewer 1 Report

This paper reviews different approaches based on reprogramming to generate cardiomyocytes. The authors presented a brief introduction, described the main challenges, and discussed some clinical aspects.

The paper is exciting for the journal readers, and I would like to note a couple of issues:

  1. Line 47: change number "4" with "four". Please review the paper for more cases of this issue.
  2. Line 55: "differentiation such as atrial" -> "differentiation, such as atrial"
  3. Line 57: "of ventricular subtype" change to "of the ventricular subtype".
  4. Line 58: "indicating possible" change to "indicating the possible" or "indicating a possible".
  5. Line 71: Perhaps, instead of "A number of", you could start the sentence with Some or Many...
  6. Line 125 to 129: Please, rewrite the sentence because it is difficult to read.
  7. Figure 1: Increase the size of the texts as it is hard to read.
  8. Line 225: Change "condition" per "conditions".
  9. In general, the authors must review some terms like "Bmp4" -> BMP4 or "Gsk3" -> GSK-3. Please, use the right abbreviations. There are many other examples.

Author Response

We appreciate the excellent suggestions to improve our manuscript and have gone point by point to address each issue pointed out by the reviewer. Our revised document contains revisions that account for all 9 comments and hope that the manuscript is now up to your standards for publication.

Reviewer 2 Report

Jiang et al. have written a comprehensive review about the strategies and challenges to improve cellular programming-based approaches for heart generation therapy. This review is well written and addresses adequately concerns regarding the clinical applications and use of iPSC-based therapies. I warmly support publishing this excellent review.

Author Response

We appreciate the excellent suggestions to improve our manuscript and have gone point by point to address each issue pointed out by the reviewer. Our revised document contains revisions that account for all 4 comments and hope that the manuscript is now up to your standards for publication.

Reviewer 3 Report

Jiang and colleagues describe the reprogramming-based approaches for generating cardiomyocytes are briefly introduced and reviewed and the challenges, including low efficiency, functional maturity, and safety issues that hinder further translation of these approaches into a clinical setting are discussed. It is a well-written review manuscript interesting to read, and some minor comments need to be addressed for further consideration.

Comments:

  1. In the title, it will be more appropriate to change the word ‘generation’ into ‘regeneration’.
  2. In many places the sentences are more complex and lengthier. Please make it simple sentences.
  3. Please rewrite the sentence in line 169-171. In this sentence, I could see the word ‘for’ used in four times.
  4. In the next sentence line 171, insert “in’ in between the words ‘difficulty recapitulating’ and reads as ‘difficulty in recapitulating’.

Author Response

We really appreciate your effort in helping to prepare and improve our manuscript. We hope that the points we have addressed from other reviewers have made our manuscript better and satisfactory for publication.